# Experimental implementation of fully controlled dephasing dynamics and synthetic spectral densities

Zhao-Di Liu[1,2], Henri Lyyra[3], Yong-Nan Sun[1,2], Bi-Heng Liu [1,2], Chuan-Feng Li[1,2], Guang-Can Guo[1,2], Sabrina Maniscalco[3,4] & Jyrki Piilo [3]

Engineering, controlling, and simulating quantum dynamics is a strenuous task. However, these techniques are crucial to develop quantum technologies, preserve quantum properties, and engineer decoherence. Earlier results have demonstrated reservoir engineering, construction of a quantum simulator for Markovian open systems, and controlled transition from Markovian to non-Markovian regime. Dephasing is an ubiquitous mechanism to degrade the performance of quantum computers. However, all-purpose quantum simulator for generic dephasing is still missing. Here, we demonstrate full experimental control of dephasing allowing us to implement arbitrary decoherence dynamics of a qubit. As examples, we use a photon to simulate the dynamics of a qubit coupled to an Ising chain in a transverse field and also demonstrate a simulation of nonpositive dynamical map. Our platform opens the possibility to simulate dephasing of any physical system and study fundamental questions on open quantum systems.

[1] CAS Key Laboratory of Quantum Information, University of Science and Technology of China, Hefei 230026, China. [2] Synergetic Innovation Center of Quantum Information and Quantum Physics, University of Science and Technology of China, Hefei 230026, People's Republic of China. [3] Turku Centre for Quantum Physics, Department of Physics and Astronomy, University of Turku, FI-20014 Turun yliopisto, Finland. [4] Centre for Quantum Engineering, Department of Applied Physics, Aalto University, FI-00076 Aalto, Finland. These authors contributed equally: Zhao-Di Liu, Henri Lyyra. Correspondence and requests for materials should be addressed to C.-F.L. (email: cfli@ustc.edu.cn) or to J.P. (email: jyrki.piilo@utu.fi)

 1

When a quantum system of interest interacts with an environment, its evolution becomes nonunitary and displays decoherence[1]. This loss of quantum properties is interesting in itself for fundamental aspects—such as quantum-to-classical transition[2]—but it is also important when developing applications of quantum physics for technological purposes[3]. Therefore, the dynamics of open quantum systems has become a major research area in modern quantum physics incorporating a multitude of physical systems and platforms.

Since it is hard, or even impossible, to avoid decoherence in realistic quantum systems, it is important to find means to control noise, and to develop new theoretical and simulation tools for open quantum systems. Indeed, already quite some time ago reservoir engineering was demonstrated experimentally with trapped ions by applying noise to trap electrodes[4], and thereby also influencing how the open system evolves. It is also possible to monitor in time the decoherence of field states in a cavity[5]. More recently, a quantum simulator for Lindblad or Markovian dynamics was constructed, motivated by the studies of open many-body systems[6,7], and a simulator for noise induced by fluctuating fields was introduced in ref. [8]. There has also been a large amount of activities dealing with non-Markovian quantum dynamics[9–11], including an experiment for controlled Markovian to non-Markovian transition with dephasing in a photonic system[12], and others induced by similar motivations[13,14].

We focus on dephasing, or pure decoherence, which is an ubiquitous mechanism leading to a loss of quantum properties and degrading the performance of quantum computers[15]. Indeed, dephasing appears naturally in multiple physical systems and processes, including qubit coupled to harmonic oscillators in thermal equilibrium[1], central spin coupled to Ising chain in transverse field[16], excitons in quantum dots[17,18], superconducting qubits influenced by fluctuating magnetic dipoles[19], and particles in a spatial superposition in gravitational field[20,21]—to name few examples.

However, despite of all the earlier theoretical and technological progress, full experimental control of decoherence—allowing to emulate arbitrary open system dephasing dynamics—has turned out to be an elusive goal. Having a complete freedom to induce any nonunitary dynamics for a given system in the laboratory would allow to simulate complex dynamical phenomena from a wide variety of fields, e.g., spin systems. This would also allow one to find out what are the ultimate limits of decoherence control. Here, we implement arbitrary and fully controlled dephasing dynamics in the laboratory, which opens also the prospect to simulate open system qubit dynamics essentially in any physical system including those mentioned above. Moreover, our results demonstrate that it is possible to induce decoherence patterns that are not produced by ambient reservoirs and their spectral densities, i.e., to manufacture artificial, or synthetic, spectral densities. On the most fundamental level, full control of open system dynamics allows the simulation of dynamical maps that are not completely positive or positive. These concepts and the problematics of appropriate properties of dynamical maps have been extensively debated in the open system theory for long time[22–24].

## Results

**Theoretical description of dephasing control.** Our goal is to control and simulate dynamical maps described by a family of $t$-parametrized pure dephasing channels $\Lambda_t$ such that

$$\Lambda_t(\rho(0)) =: \rho(t) = \begin{pmatrix} \rho_{00} & D^*(t)\rho_{01} \\ D(t)\rho_{10} & \rho_{11} \end{pmatrix}. \quad (1)$$

Here, $\rho_{ij}$ ($i, j = 0, 1$) are the elements of the qubit density matrix $\rho$

at initial time $t = 0$ and $D(t) \in \mathbb{C}$ is the so-called decoherence function. In dephasing, the diagonal elements $\rho_{00}$ and $\rho_{11}$ corresponding to populations do not evolve whereas $D(t)$ contains information on how the coherences $\rho_{01}$ and $\rho_{10}$ of the qubit evolve. If $|D(t)|$ decreases monotonically, so does also the magnitude of coherences. However, to develop a generic simulator for dephasing, we need to implement arbitrary $|D(t)|$, and subsequent evolution of the magnitude of coherences, without influencing the populations.

The open system qubit in our simulator is the polarization of a photon and the environment consists of its frequency degree of freedom. The scheme is based on full control over the total initial polarization–frequency state, which then dictates the subsequent polarization dephasing dynamics when the interaction between the polarization and frequency degrees of freedom—and the open system time evolution—begins. An initial pure polarization–frequency state for the photon can be written as

$$|\Psi\rangle = C_V|V\rangle \int g(\omega)|\omega\rangle d\omega + C_H|H\rangle \int e^{i\theta(\omega)} g(\omega)|\omega\rangle d\omega. \quad (2)$$

Here, $V$ ($H$) corresponds to vertical (horizontal) polarization with amplitude $C_V$ ($C_H$), $\omega$ are the frequency values with amplitude $g(\omega)$, and $\theta(\omega)$ is the frequency dependent phase factor for polarization component $H$. The probabilities are normalized in the usual manner with $|C_H|^2 + |C_V|^2 = 1$ and $\int |g(\omega)|^2 d\omega = 1$. It is important to note here that having a limited control over the initial frequency distribution $P(\omega) = |g(\omega)|^2$, e.g., implementing double peak structure, allows some degree of engineering of the dephasing dynamics[12]. However, for generic simulator we need full control over both the frequency distribution and the frequency–polarization dependent phase distribution $\theta(\omega)$. This also means that we are exploiting in our simulator initial polarization–frequency correlations which happens as soon as we have nonconstant distribution for $\theta(\omega)$. In this case, the initial state Eq. (2) can not be written as a polarization–frequency product state.

Once the initial state given by Eq. (2) has been prepared, the simulator dynamics occurs when polarization and frequency interact in birefringent medium, such as quartz or calcite. The evolution of the total state is governed by the Hamiltonian

$$H = (n_H|H\rangle\langle H| + n_V|V\rangle\langle V|) \int 2\pi\omega|\omega\rangle\langle\omega| d\omega, \quad (3)$$

where $n_H$ ($n_V$) is the refractive index of the medium in the direction $H$ ($V$). By tracing out from the total system evolution the frequency degree of freedom, the polarization state undergoes the following dephasing dynamics:

$$\rho(t) = \begin{pmatrix} |C_H|^2 & \kappa^*(t)C_H C_V^* \\ \kappa(t)C_V C_H^* & |C_V|^2 \end{pmatrix}, \quad (4)$$

where

$$\kappa(t) = \int |g(\omega)|^2 e^{i\theta(\omega)} e^{i2\pi\Delta n\omega t} d\omega, \quad (5)$$

$\Delta n = n_H - n_V$ and $t$ is the interaction time. Eq. (5) shows in a clear manner that the decoherence function $\kappa(t)$ is the Fourier transformation of the distribution $|g(\omega)|^2 e^{i\theta(\omega)}$ used to prepare the tailored initial total system state. Since Fourier transform is invertible, this connection tells us how the distributions $g(\omega)$ and $\theta(\omega)$ should be chosen to induce any desired polarization dephasing dynamics defined by any complex function $\kappa(t)$. On

the other hand, for each $\kappa(t)$ the corresponding complex distribution $|g(\omega)|^2 e^{i\theta(\omega)}$ is unique, and thus the implementation of a nontrivial $\theta(\omega)$ is necessary for full freedom of choosing the dephasing dynamics. For generic open quantum systems, specifying the spectral density (i.e., the coupling with the environment) is not equivalent to specifying the analytic expression of the dynamical map (solution of the master equation). In fact, in general, one may not even be able to solve analytically the master equation, and have a closed analytical form of the dynamical map. However, the case of pure dephasing dynamics is different because specifying the spectral density uniquely fixes the analytical form of the solution since decoherence function (off-diagonal term of the density matrix) only depends on the spectral density, see Eqs. (2), (4), and (5).

The complete positivity and positivity (P) conditions for single-qubit dephasing channel in Eq. (1) are the same, namely $|D(t)| \leq 1$. However, the versatility of our simulator and control over $\kappa(t)$ permit to simulate nonpositive channels in the following way. Due to initial system–environment correlations induced by the nontrivial distribution $\theta(\omega)$, we have cases with $|\kappa(0)| < 1$, i.e., the simulator uses restricted domain of initial polarization states. To simulate the channel in Eq. (1) and its decoherence function $D(t)$ with the simulator function $\kappa(t)$ in Eq. (5), we need to use the scaling $|D(t)| = |\kappa(t)|/|\kappa(0)|$. Therefore, with the full control of the simulator and using the initial system–environment correlations, we can also generate dynamics with $|\kappa(t)| > |\kappa(0)|$, i.e., $|D(t)| = |\kappa(t)|/|\kappa(0)| > 1$, and hence can simulate also nonpositive maps.

**Experimental setup**. In the experiment, a photon pair is produced in spontaneous parametric down conversion (SPDC) process by pumping a type-II beta-barium-borate crystal ($9.0 \times 7.0 \times 1.0$ mm$^3$, $\theta = 41.44°$) by a frequency-doubled femtosecond pulse (400 nm, 76 MHz repetition rate) from a mode-locked Ti: sapphire laser. After passing through the interference filter (3 nm FWHM, centered at 800 nm), the photon pairs are coupled into single-mode fibers separately. One of the photons is sent to the experimental device described in Fig. 1, and the other is used as a trigger for data collection. The coincidence counting rate

collected by the avalanche photo diodes (APDs) is about $1.8 \times 10^5$ in 60 s and the measurement time for each experiment was 10 s.

In the device of Fig. 1, a half-wave plate (HWP) is used to maximize the $H$ polarized component and a polarizing beam splitter (PBS) completely filters out the $V$ polarized component of the photon. Another HWP rotates the polarization from $|H\rangle$ to balanced superpositions $\frac{1}{\sqrt{2}}(|H\rangle \pm |V\rangle)$. A beam displacer displaces the $V$ polarized component to lower branch, allowing us to manipulate the polarization components independently. Before the photon goes through three gratings, the $V$ polarized component goes through the HWP core of the composite component glass cemented half-wave plate (GCHWP) and gets rotated to $H$. This is to avoid errors caused by the polarization dependency of the grating efficiency and the ability of the SLM to modulate only the $H$ polarization.

Then the photon is diffracted in the horizontal direction with three cascaded gratings (1200 l/mm), and thus the frequency modes are converted into spatial modes. A collimating lens (PCCL) transforms the spatial modes into parallel lights incident on the phase-only spatial light modulator (SLM). In our work, we need to implement dephasing dynamics shown in Fig. 2. This is achieved by engineering the photon frequency and phase distribution displayed in Fig. 3. For the latter, the SLM can introduce complex phase factors for the spatial modes [Fig. 3 (e–h)]. In order to tune the intensity distribution of the frequency, gratings (25 l/mm, with parallel horizontal lines) are written in the hologram of SLM [Fig. 1, inset]. The horizontal profile (pixel number in every column) of the gratings of the hologram (GH) is designed the same as the frequency intensity distribution [Fig. 3a–d], which ensures that the area of the GH is proportional to the intensity of the frequency. If photons are not incident on the GH, they will be reflected by the screen of the SLM directly. On the other hand if photons cover the GH, they will be diffracted vertically at the first order of the GH with a fixed efficiency about 60%. By collecting only the photons diffracted at the first order of the GH we achieve the intensity modulation of frequency. These manipulations can be performed independently for the upper and lower branches. For simplicity, we choose the

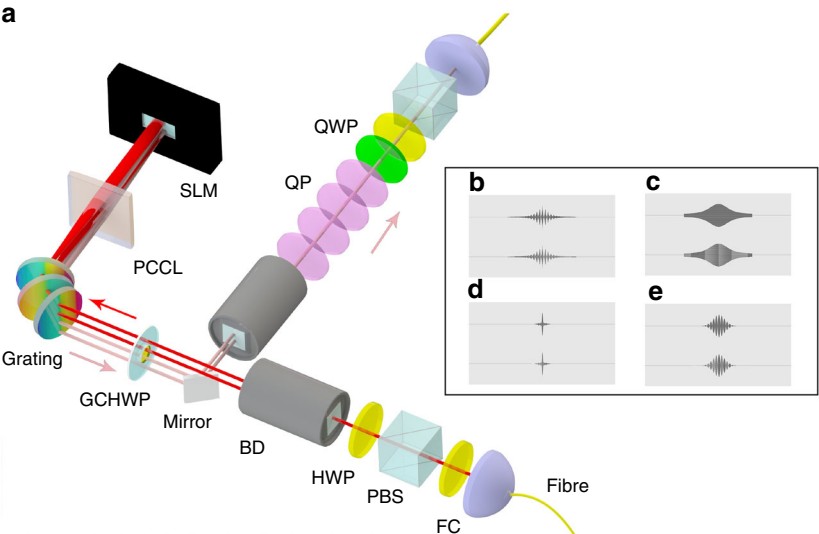

**Fig. 1** The experimental setup. **a** Key to the components: FC—fiber connector, PBS—polarizing beam splitter, HWP—half-wave plate, BD—beam displacer, GCHWP—glass cemented half-wave plate, PCCL—plano convex cylindrical lense, SLM—spatial light modulator, QP—quartz plate, and QWP—quarter-wave plate. The photon is guided from the source to the device via the lower FC. Then the photon goes through the gratings (the dark red lines) to SLM where the state is manipulated and the photon is reflected back (light red lines). A mirror guides the returning photon through the quartz plate combination. Finally, a combination of QWP, HWP, and PBS is used to run tomographic measurement at the end of the device. **b**–**e** The holograms used in the experiment so that (b-e) correspond to Fig. 3(a-d)

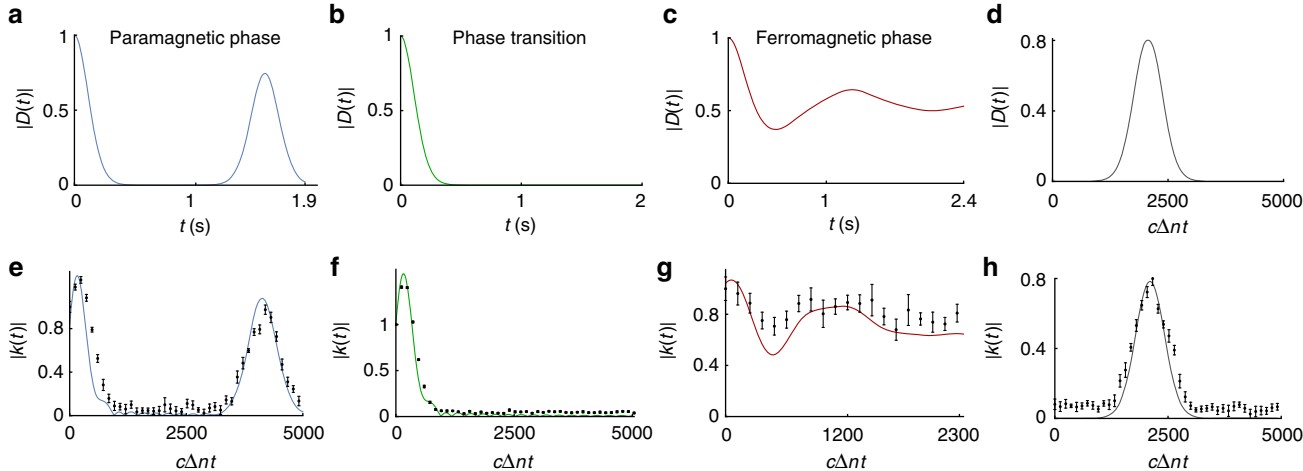

**Fig. 2** Model decoherence functions and their simulation. **a–c** The dynamics of the decoherence function $D(t)$ in the spin–Ising chain model as function of time in seconds. a–c correspond to $\lambda = 0.01$, $\lambda = 0.9$, and $\lambda = 1.8$, respectively. **d** The dynamics of the decoherence function $D(t)$ for the dephasing channel Eq. (1) for the case when positivity is broken. **e–g** Experimental dynamics of the decoherence function $|\kappa(t)|$ in the simulator corresponding to panels (a–c) as function of effective path difference in units of 800 nm. The black dots correspond to measurement data and the error bars are mainly due to the counting statistics, which are standard deviations calculated by the Monte-Carlo method. The solid curves are theoretical fits for the measurement data which have been obtained by using the width of the photon frequency window as fitting parameter. **h** The dynamics of the decoherence function $|\kappa(t)|$ when simulating nonpositive map of panel (d). The results clearly display the dynamical property $|\kappa(t)| > |\kappa(0)|$ over a long interval of time

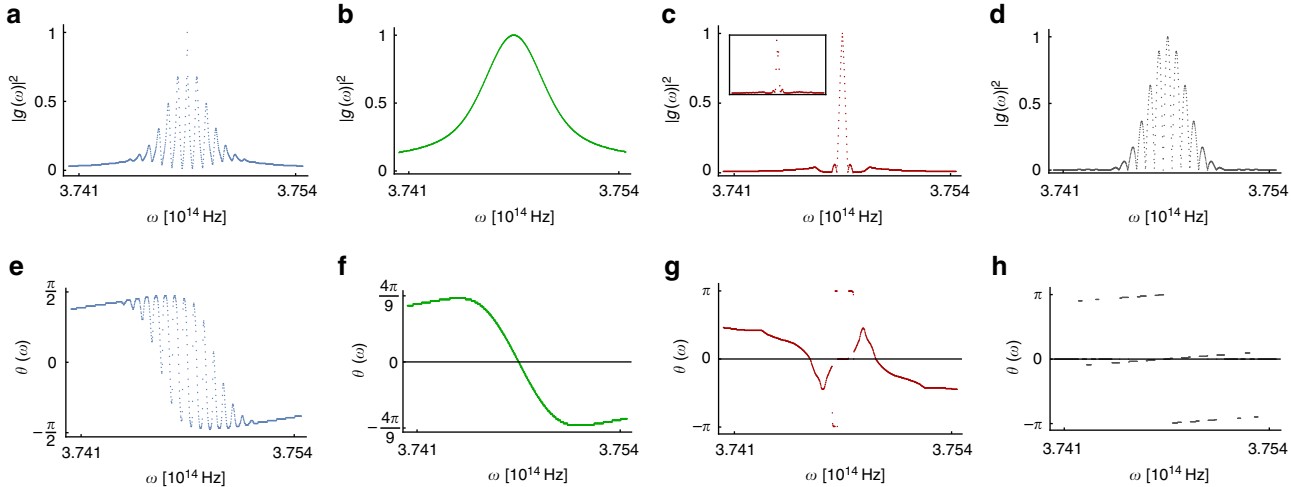

**Fig. 3** Implemented photon frequency and phase distributions. **a–d** Probability distributions $|g(\omega)|^2$ of the photon frequency. **a–c** correspond to spin–Ising model simulation and (**d**) corresponds to nonpositive map simulation. Inset in panel (**c**) shows the experimentally measured distribution for this case after the SLM implementation. **e–h** Phase distributions $\theta(\omega)$ of the photon frequency corresponding to (**a–d**). These distributions were implemented pairwise in the experiment with a two-dimensional SLM using the effective resolution of 900 pixels in frequency modes. The bandwidth of each distribution is approximately 3 nm and they are centered at 800 nm. The measurement data of the simulator dynamics corresponding to implemented distribution pairs (a, e), (b, f), (c, g), and (d, h) are shown in Fig. 2(e–h), respectively

reflected intensity distribution to be the same for both branches and implement the complex phase distributions on the lower branch only.

From the SLM the photon is reflected back (the light red lines in Fig. 1) through the PCCL and three gratings, which collimate and combine the spatial modes into one mode in each branch. The branches go again through the GCHWP, which rotates this time the polarization of the upper branch from *H* to *V*. A mirror guides both branches through another BD, which recombines them into one. This way we have prepared the total polarization–frequency state in the shape of Eq. (2). Controlling the reflected intensity and complex phase distributions with

SLM this way gives us directly full freedom to implement the distributions $g(\omega)$ and $\theta(\omega)$ in Eq. (2), respectively. Thus, the setup gives us full control of the dephasing dynamics of the polarization state as shown in Eq. (5). Note that SLMs have been recently used also for quantum computing and information purposes, see, e.g., refs. [25–27], and that a 4f-line is a standard way to manipulate the spectrum and implement pulse shaping by optical means[28]. Finally, the recombined photon goes through a combination of quartz plates (QP), which couple the polarization with frequency according to interaction Hamiltonian (3). The total interaction time is controlled by changing the thickness of the QP combination. For each selected interaction time *t*, a

combination of a quarter-wave plate, HWP, and PBS is used to run a tomographic measurement to determine the density matrix $\rho(t)$ of the polarization qubit.

**Using a photon to simulate qubit coupled to Ising chain.** To give an experimental demonstration of our optical simulator, we focus on the dynamics of an open system qubit interacting with an environment whose ground state exhibits a quantum phase transition. We consider a central spin coupled to an Ising spin chain in a transverse field. This is a widely used complex spin interaction model[16,29] where one can induce the ground state phase transition by changing the magnitude of the transverse magnetic field with respect to the Ising chain spin–spin coupling. It is also worth mentioning that for this model, by quantifying the non-Markovianity of the central spin dynamics, one can identify the point of the phase transition in the environment[29].

The dynamics of the total spin–chain system is described by the Hamiltonian[16]

$$\mathrm{H}(J, \lambda, \delta) = -J\Sigma_j \left( \sigma_3^{(j)}\sigma_3^{(j+1)} + \lambda\sigma_1^{(j)} + \delta|e\rangle\langle e|\sigma_1^{(j)} \right),$$

where $J$, $\delta$, and $\lambda$ correspond to the strengths of the nearest neighbor coupling in the chain, the spin–chain coupling, and the transverse field, respectively, while $\sigma_1$ and $\sigma_3$ are the Pauli spin operators. When the Ising chain is initially in the ground state of the environmental Hamiltonian, the dynamics of the central spin is described by the dynamical map in Eq. (1), where the time-dependent decoherence function becomes

$$D(t) = \Pi_{k>0}(1 - \sin^2(2\alpha_k)\sin^2(\varepsilon_k t)).$$

Here, $\varepsilon_k$ are the single quasiexcitation energies, and $\alpha_k$ are Bogoliubov angles, both of which depend on $\lambda$[16].

What makes this model especially interesting to simulate, is the variety of the dynamics it can induce. Specifically, fixing the parameters $J = 1$ and $\delta = 0.1$, different choices of $\lambda$ lead to very different behaviors. Figure 2a–c displays the dynamics of the decoherence function for parameters $\lambda = 0.01$, $\lambda = 0.9$, and $\lambda = 1.8$, using 4000 spins in the environment. Here, $\lambda = 0.01$ ($\lambda = 1.8$) corresponds to paramagnetic (ferromagnetic) phase of the environment and the phase transition between the two happens at $\lambda = 0.9$. When the enviroment is in the paramagnetic phase, the decoherence function in Fig. 2a decreases quite quickly destroying the coherences which, however, revive after a long-time interval, displaying also non-Markovian effects. At the phase transition point, corresponding to Fig. 2b, coherences quickly decay. In the ferromagnetic phase of the environment, the magnitude of coherences oscillates and displays trapping (see Fig. 2c).

To simulate the dephasing dynamics displayed in Fig. 2a–c, we use the inverse of the transformation in Eq. (5) to obtain the distributions $|g(\omega)|^2$ and $\theta(\omega)$, which need to be experimentally realized to prepare the appropriate initial total state of the simulator. The corresponding distributions for $|g(\omega)|^2$ are shown in Fig. 3a–c and for $\theta(\omega)$ in Fig. 3e–g. We have prepared and used initial values $C_H = 1/\sqrt{2}$ and $C_V = \pm 1/\sqrt{2}$ for polarization in the initial states of Eq. (2). The values of $|\kappa(t)|$ during the evolution are obtained via state tomography and by using trace distance[12]. The experimental results for the dephasing dynamics are displayed in Fig. 2e–g. By comparing the spin–Ising model dephasing dynamics of Fig. 2a–c to their experimental simulation in Fig. 2e–g, we observe that the simulator produces faithfully the dynamics of the central spin in the Ising model for both different phases of the environmental ground state as well as at the phase transition point. Our results demonstrate high-level of control

and versatility of the simulator and, in the considered exemplary cases, the ability to emulate dephasing in three distinct dynamical regimes: fast decoherence with revival of coherences (paramagnetic environment), fast and monotonic loss of coherences (phase transition of the environment), and coherence oscillations with trapping (ferromagnetic environment).

It is worth noting that systematic errors have a nonnegligible effect. Figure 2g is a typical example of the resolution that leads to these errors. The corresponding hologram is in Fig. 1d . It becomes more difficult to modulate the amplitude of the frequency with high fidelity, when the spectrum gets narrower (also beam mode becomes worse). Although the setup was carefully optimized, these factors still lead to decrease in the fidelity of the initial state preparation. This is the reason why the experimental result [Fig. 2g] does not show an agreement with the theory [Fig. 2c] as good as in the other figures.

Having too wide spectrum also leads to systematic errors. Figure 2a–c display the dynamics of the decoherence function for the spin–system parameters $\lambda = 0.01$, $\lambda = 0.9$, and $\lambda = 1.8$, respectively, using 4000 spins in the environment. Although three gratings (1200 l/mm) are used, 3 nm FWHM (full width at half maximum) of SPDC photons can only cover 900 pixels in SLM. This effectively means that we can simulate 900 out of all 4000 environmental spins. Totally, 3100 spins corresponding to amplitude and phase close to 0 are ignored in the setup. The experimental results [Fig. 2e–g] are slightly different with respect to the theoretical results [Fig. 2a–c]. In principle, this systematic error can be reduced by using smaller pixel SLM and wider FWHM filter. Note that, unless the spectrum is too narrow, the beam mode is good enough to achieve simulation with high fidelity.

For gratings, three 1200 l/mm gratings are used in our setup. This is a result of tradeoff. Totally, 1800 l/mm grating will make the divergence of spectrum bigger, but the fidelity of polarization states will be significantly less than the fidelity with gratings of 1200 l/mm. Therefore, increasing the divergence angle of the spectrum by increasing the density of the grating is somewhat challenging.

It is evident from Fig. 3, that producing this high control of dephasing, and being able to simulate a wide variety of dynamical features, indeed requires very challenging control of the photon frequency and frequency dependent phase distribution. This precision is exactly what allows for the accurate mimicking of the dynamics in different dephasing regimes, even though the number of pixels of the SLM is large but still limited.

**Implementing nonpositive dynamical map.** To demonstrate the ability to simulate a non-positive dynamical map, we choose the decoherence function dynamics displayed in Fig. 2d for the map of Eq. (1). This requires implementing initial frequency and phase distributions shown in Fig. 3d, h, respectively. Comparing the decoherence function of Fig. 2d to the experimental simulation in Fig. 2h, we observe again the accuracy and power of the simulator. Therefore, our results give a proof-of-principle demonstration that, with our scheme, it is possible to simulate a class of dynamical maps, which breaks a property traditionally considered as the ultimate criterion for discriminating between the type of open system dynamics that could occur naturally (or be engineered) and those which were considered unphysical. It is also interesting to note here, that the initial frequency distribution to simulate the paramagnetic phase of the spin–Ising model [Fig. 3a] is very similar to the distribution used to simulate nonpositive map [Fig. 3d]—even though the dynamics is quite different (see Fig. 2e, h). The difference between the two arises from the completely different type of phase distributions $\theta(\omega)$, shown in

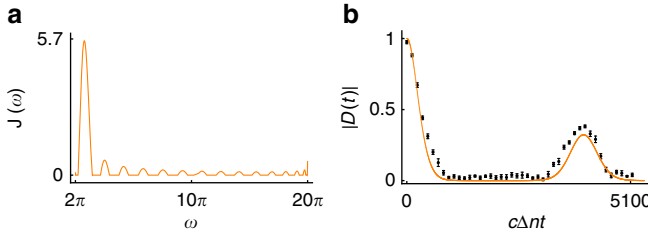

**Fig. 4** Synthetic spectral density and corresponding decoherence dynamics of a qubit. **a** A chosen spectral density $J(\omega)$ used in Eq. (6). The unit of frequency is $\lambda_0/c$ with $\lambda_0 = 800$ nm. **b** Corresponding theoretical and experimental dynamics of the decoherence function (for more details, see the main text). The black dots correspond to measurement data and the error bars are mainly due to the counting statistics, which are standard deviations calculated by the Monte-Carlo method. The evolution time is given by effective path difference in units of $\lambda_0$

Fig. 3e, h for the two cases. This again reflects the crucial role that $\theta(\omega)$ plays in developing and implementing generic simulator for dephasing.

**Synthetic spectral densities and other extensions**. It is worth noting that our results makes it possible to produce synthetic spectral densities when considering the environments that an open system interacts with. Consider, for example, a qubit interacting with a bosonic environment via the interaction $H_i = \sum_k \sigma_3 (g_k a_k + g_k^* a^\dagger)$, where $\sigma_3$ is the qubit Pauli operator, $g_k$ the coupling constant to the bosonic mode $k$, and $a_k$ ($a_k^\dagger$) the creation (annihilation) operator for mode $k$. Then the decoherence function can be written as

$$D(t) = \exp\left[-\int_0^\infty d\omega J(\omega) \coth\left(\frac{\beta\omega}{2}\right) \frac{1-\cos(\omega t)}{\omega^2}\right], \quad (6)$$

where $\beta$ is the inverse temperature and $J(\omega)$ is spectral density, which contains information about the properties of the environment[1]. Therefore, having a simulator for any behavior of $D(t)$, allows us via the connection above also to simulate generic spectral densities $J(\omega)$. This means that also open system dynamics and spectral densities which do not appear in nature otherwise, i.e., synthetic spectral densities, can be created. Ohmic spectral density is often used for dephasing dynamics. However, as an example of synthetic spectral density, we choose the one shown in Fig. 4a, which produces decoherence dynamics displayed both theoretically and experimentally in Fig. 4b. The theoretical dynamics is obtained numerically from Eq. (6) by using zero-temperature environment and the $J(\omega)$ displayed in Fig. 4a. The experimental result has been obtained by using the frequency distribution $|g(\omega)|^2$ of the simulator photon displayed in Fig. 3a, while the used initial phase distribution $\theta(\omega)$ is, instead of Fig. 3e, a constant function. This result gives experimental evidence on the realizability of arbitrary synthetic spectral densities which we plan to study in more detail in the future.

Current framework can be extended to multiqubit case by using the presented dephasing engineering scheme for each of the qubits. By using both of the SPDC photons, also initial correlations between the environments (frequencies) of the qubits can be controlled to a certain extent allowing to combine dephasing control with nonlocal features of the dynamical map[30,31]. Moreover, it is also possible to include coherent operations to the existing setup allowing for the exploitation of the combination of sequences of coherent operations and controllable decoherent operations, in a many-body scenario, for quantum control and simulation purposes[6,7]. Lastly, activities using structured light have increased significantly during the

recent years[32], including also photonic experiments[33]. Here, our results open the possibility to combine interferometry with fully controllable noise and structured photons.

## Conclusions

Summarizing, we have introduced and realized experimentally a generic simulator for one-qubit dephasing. The features of the simulator include full control of the dephasing, therefore allowing us in principle to simulate any pure-decoherence dynamics. As examples we considered dephasing for an Ising model in a transverse field, where the environment exhibits a phase transition, and the dynamics of the qubit displays three distinct and highly-different features. Moreover, we also showed how to simulate a nonpositive dynamical map. The ability to synthesize arbitrary dephasing dynamics establishes an experimental testbed for fundamental studies on long-debated but not yet settled questions. In general, our results have implications in all fields and physical contexts, where dephasing plays a key role. These include, among others, quantum probing of many-body systems, exciton transfer in light-harvesting complexes, and numerous experimental platforms for quantum technologies.

**Data availability.** The data that support the findings of this study are available from the authors upon reasonable request.

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

## Acknowledgments

The Hefei group was supported by the National Key Research and Development Program of China (No. 2017YFA0304100), the National Natural Science Foundation of China (Nos. 61327901,11774335), Key Research Program of Frontier Sciences, CAS (No. QYZDY-SSW-SLH003), the Fundamental Research Funds for the Central Universities (No. WK2470000026), and Anhui Initiative in Quantum Information Technologies (AHY020100). The Turku group acknowledges the financial support from Horizon 2020 EU collaborative project QuProCS (Grant Agreement 641277), Magnus Ehrnrooth Foundation, and the Academy of Finland (Project no. 287750). H.L. acknowledges also the financial support from the University of Turku Graduate School (UTUGS).

## Author contributions

Z.-D.L., Y.-N.S, B.-H.L. C.-F.L., and G.-C.G. planned, designed, and implemented the experiments under the supervision of C.-F.L. and G.-C.G. Most of the theoretical analysis was peformed by H.L. under the supervision of S.M. and J.P. The paper was written by Z.-D.L., H.L., C.-F.L., S.M., and J.P., while all authors discussed the contents.

## Additional information

**Competing interests:** The authors declare no competing interests.

