## [Peer Review File · Nature Communications]

Reviewers' comments:

Reviewer #1 (Remarks to the Author):

The manuscript concerns the theoretical and experimental investigation on the engineering of a fully general dephasing dynamics of an open quantum system. In particular, the authors consider the polarization degrees of freedom of one of the two photons produced via spontaneous parametric down conversion (SPDC) as the open system, and the corresponding frequency degrees of freedom as the environment. Given a general pure polarization-frequency initial state as in equation (2) and the evolution fixed by the total Hamiltonian in equation (3), the resulting polarization open-system dynamics will be given by the pure dephasing in equation (4), fully determined by the function $k(t)$ in equation (5); thus, by properly choosing the initial amplitude $g(\omega)$ and frequency-dependent phase $\theta(\omega)$, one can reproduce any pure dephasing dynamics. Especially, the authors focus their attention on two different kinds of dynamics: one referred to a spin-chain system, in which the environmental chain is in two different phases (paramagnetic and ferromagnetic) depending on the value of the parameters of its free Hamiltonian, and the second where the system qubit undergoes a "non-positive evolution" due to the presence of initial correlations between the system and the environment.

From an experimental point of view, the authors exploit an all-optical setup, where, as said, the global system under investigation is given by one SPDC photon (the other is used for coincidence counting) and the time-evolution is given by the passage through a birefringent crystal. The most demanding part of the experiment is the preparation of the initial state, which involves 3 gratings to convert back and forth the frequency degrees of freedom into spatial ones and, crucially, a spatial light modulator which can introduce the (practically) arbitrary phase functions needed to reproduce a general dephasing dynamics; at the same time, the amplitude distribution is controlled via gratings written in the hologram of the SLM. Importantly, possible delays due to the polarization dependency of the grating efficiency and of the SLM are avoided by means of a proper glass cemented half-wave plate.

The ability to control the dynamics of open quantum systems is a crucial step toward the actual employment of quantum features, e.g., in view of the development of quantum technologies. In particular, the possibility to engineer a controlled environment (or part of it) to determine the resulting open-system evolution is a topic which has been attracting more and more interest, due to the increased experimental capabilities, which has also induced to go well beyond the "standard" Markovian characterization of open-system dynamics.

The results reported by the authors in the present manuscript certainly provide a significant step forward in the control and engineering of open-quantum-system dynamics. The capability to reproduce in a controlled way practically any pure dephasing dynamics is in fact of crucial interest, both because of the generality of the dynamics which can be engineered (which significantly overcomes the results reported so far in the literature) and because of the ubiquitous and fundamental nature of pure dephasing. Accordingly, the experimental effort has clearly increased with respect to previous settings, as required by the preparation of the general initial photon state needed to reproduce any pure dephasing dynamics. In addition, let me also mention that the manuscript is clearly written and both the theoretical and experimental analysis are plainly explained, fully supporting the claims of the authors.

Thus, I am happy to recommend the paper for publication in Nature Communications; before publication, I would like the authors to address the following (minor) remarks.

1) I find misleading the statement at page 3, second column that $k(t)$ is independent of the used initial qubit state. Of course, it is independent from the polarization amplitudes CH and CV , but in general, due to the presence of initial system-environment correlations, this will not be the same as being independent from the initial qubit state. In fact, the latter is characterized by the initial off-diagonal element $\rho_{10}(0) = k(0)CH^*CV$ (and the complex conjugate element): $k(0)$ and then $k(t)$ are related with the initial qubit state (note that $k(0) = 1$ by definition of $g(\omega)$ if there are no initial

correlations, i.e., $\theta(\omega)=0$).

This is a rather important point, especially with reference to the meaning of non-positive maps. As known in the literature (see, e.g., the references [15-17]) these are still a perfectly legitimate mathematical tool to describe open system dynamics, since due to the presence of initial system-environment correlations they are defined in a physically meaningful way only on a restricted domain of possible initial states, where the action of the map is guaranteed to be well defined (i.e. to map those specific initial states into positive states at any time). In the example reported by the authors, this is precisely reflected by the connection between the initial off-diagonal element and $k(0)$: for fixed $g(\omega)$ and $\theta(\omega)$, $k(0)$ will be fixed and then the set of allowed initial states will be the set of all states of the form as in equation (4) for $t=0$, for different choices of CH and CV (with $|CH|^2 + |CV|^2=1$). Hence, e.g., for $|k(0)|<1$ the set of allowed states will be an ellipsoid strictly included in the Bloch sphere, so that the physical meaning of the map will be guaranteed within this ellipsoid and not outside it. (The dynamics in Fig.2 d) represents a very nice example of this, with a minimal size of the ellipsoid of initial well-defined states).

2) I think that something more specific should be said about the possibility to extend the present platform to the multi-qubit case, also due to the comparison with other settings used in the literature (see, e.g., [6,7]). The authors briefly mention something about this at the end of the first column of page 6, but this seems to refer only to the "trivial" situation where each qubit is treated independently. Could it be possible to do something also considering entangled qubits, possibly even simulating an interaction among them? For example, could one exploit for this both the photons generated by SPDC?

Reviewer #2 (Remarks to the Author):

Report for manuscript NCOMMS-18-08713, "Experimental implementation of fully controlled dephasing dynamics and synthetic spectral densities".

In the manuscript NCOMMS-18-08713, the authors report on the theoretical- experimental demonstration of an optical quantum simulator of single-qubit dynamics under the influence of pure dephasing. As prototype system the authors consider a single photon with fully controlled polarization and frequency distributions.

In essence, the authors exploit the fact that the reduced density matrix describing a single photon in the polarization-frequency domain exhibits a modulation function in the coherence terms. Then, by comparing the single-photon density matrix to a generic single-qubit dynamical map under the influence of pure-dephasing, the authors elucidate that both matrices exhibit the same structure. As a result, the modulating function for the single-photon coherences can be mapped to the decoherence function of the qubit.

In the single-photon context, the "decoherence function" is the Fourier transform of the initial frequency distributions times a frequency-dependent-phase-factor for the horizontal polarization component. Hence, by tailoring the initial frequency distribution of the single photon the authors are able to emulate the dynamics of a single-qubit under the influence of pure dephasing. In short, I find this idea simple and elegant.

The paper is extremely interesting and well written. The theoretical and experimental approaches are consistent and the agreement in the outcomes is impressive. In my opinion the results are of general relevance, so that the manuscript match the level of excellence required by the Journal.

I only have a comment.

In the title the authors mention "implementation of synthetic spectral densities". However, only in the last paragraph of the manuscript they stress "it is possible to produce synthetic spectral

densities". I would encourage the authors to provide experimental evidence of this statement. After that, I think the manuscript would be appropriate for publication in Nature Communications.

Reviewer #3 (Remarks to the Author):

This manuscript describes the implementation of generic dephasing maps using spectral manipulation of single photons, coding polarization qubits. By imparting different spectral phases to Horizontal and Vertical components, the authors were able to replicate the effects of the coupling of a single qubit to an Ising spin chain with varying transverse field.

The work is undoubtedly interesting to specialists, but it lacks the elements of novelty that should be expected for Nature Communications. Many instances of optical quantum simulations have appeared in the last years, and, while this is an elegant demonstration, it has the same drawbacks as previous demonstrations, i.e. it is a neat way to simulate the output wavefunction, rather than directly mimicking the dynamics. While the former is still a relevant effort, the latter is more complex, hence appealing.

In addition, the experimental technique is applied for the first time to quantum simulation, however, a 4f-line it is a standard way to manipulate the spectrum by linear optical means, however much I can appreciate the ingenuity of the solution.

I therefore encourage the authors to seek publication in a prominent physics journal, which looks like a more appropriate choice.

The authors might want to consider the following points:

- the discussion on the impact of the limited resolution of the grating and SLM system on the simulation is quite terse. Can the authors comment clearly on what features can not be replicated due to this imperfection, e.g. in terms of a frequency cut-off in the noise spectrum?
- the bibliography on the use of SLMs for qubit and single-photon manipulation and, in general, on optical simulation is very limited. Please consider extensions.

We thank all the referees for careful reading of the manuscript and for their comments.

We give our point-by-point response below including the corresponding changes to the text of the manuscript. Most of the comments requesting modifications to the paper were minor ones. However, for the sake of completeness, we provide below in blue color also the full text of the reviewers.

REVIEWER #1

Referee:

The manuscript concerns the theoretical and experimental investigation on the engineering of a fully general dephasing dynamics of an open quantum system. In particular, the authors consider the polarization degrees of freedom of one of the two photons produced via spontaneous parametric down conversion (SPDC) as the open system, and the corresponding frequency degrees of freedom as the environment. Given a general pure polarization-frequency initial state as in equation (2) and the evolution fixed by the total Hamiltonian in equation (3), the resulting polarization open-system dynamics will be given by the pure dephasing in equation (4), fully determined by the function $k(t)$ in equation (5); thus, by properly choosing the initial amplitude $g(\omega)$ and frequency-dependent phase $\theta(\omega)$, one can reproduce any pure dephasing dynamics. Especially, the authors focus their attention on two different kinds of dynamics: one referred to a spin-chain system, in which the environmental chain is in two different phases (paramagnetic and ferromagnetic) depending on the value of the parameters of its free Hamiltonian, and the second where the system qubit undergoes a “non-positive evolution” due to the presence of initial correlations between the system and the environment.

From an experimental point of view, the authors exploit an all-optical setup, where, as said, the global system under investigation is given by one SPDC photon (the other is used for coincidence counting) and the time-evolution is given by the passage through a birefringent crystal. The most demanding part of the experiment is the preparation of the initial state, which involves 3 gratings to convert back and forth the frequency degrees of freedom into spatial ones and, crucially, a spatial light modulator which can introduce the (practically) arbitrary phase functions needed to reproduce a general dephasing dynamics; at the same time, the amplitude distribution is controlled via gratings written in the hologram of the SLM. Importantly, possible delays due to the polarization dependency of the grating efficiency and of the SLM are avoided by means of a proper glass cemented half-wave plate.

The ability to control the dynamics of open quantum systems is a crucial step toward the actual employment of quantum features, e.g., in view of the development of quantum technologies. In particular, the possibility to engineer a controlled environment (or part of it) to determine the resulting open-system evolution is a topic which has been attracting more and more interest, due to the increased experimental capabilities, which has also induced to go well beyond the “standard” Markovian characterization of open-system dynamics.

The results reported by the authors in the present manuscript certainly provide a significant step forward in the control and engineering of open-quantum-system dynamics. The capability to reproduce in a controlled way practically any pure dephasing dynamics is in fact of crucial interest, both because of the generality of the dynamics which can be engineered (which significantly overcomes the results reported so far in the literature) and because of the ubiquitous and fundamental nature of pure dephasing. Accordingly, the experimental effort has clearly increased with respect to previous settings, as required by the preparation of the general initial photon state needed to reproduce any pure dephasing dynamics. In addition, let me also mention that the manuscript is clearly written and both the theoretical and experimental analysis are plainly explained, fully supporting the claims of the authors.

Thus, I am happy to recommend the paper for publication in Nature Communications; before publication, I would like the authors to address the following (minor) remarks.

1) I find misleading the statement at page 3, second column that $k(t)$ is independent of the used initial qubit state. Of course, it is independent from the polarization amplitudes C_H and C_V , but in general, due to the presence of initial system-environment correlations, this will not be the same as being independent from the initial qubit state. In fact, the latter is characterized by the initial off-diagonal element $\rho_{10}(0)=k(0)C_H^*C_V$ (and the complex conjugate element): $k(0)$ and then $k(t)$ are related with the initial qubit state (note that $k(0)=1$ by definition of $g(\omega)$ if there are no initial correlations, i.e., $\theta(\omega)=0$).

This is a rather important point, especially with reference to the meaning of non-positive maps. As known in the literature (see, e.g., the references [15-17]) these are still a perfectly legitimate mathematical tool to describe open system dynamics, since due to the presence of initial system-environment correlations they are defined in a physically meaningful way only on a restricted domain of possible initial states, where the action of the map is guaranteed to be well defined (i.e. to map those specific initial states into positive states at any time). In the example reported by the authors, this is precisely reflected by the connection between the initial off-diagonal element and $k(0)$: for fixed $g(\omega)$ and $\theta(\omega)$, $k(0)$ will be fixed and then the set of allowed initial states will be the set of all states of the form as in equation (4) for $t=0$, for different choices of C_H and C_V (with $|C_H|^2 + |C_V|^2 = 1$). Hence, e.g., for $|k(0)| < 1$ the set of allowed states will be an ellipsoid strictly included in the Bloch sphere, so that the physical meaning of the map will be guaranteed within this ellipsoid and not outside it. (The dynamics in Fig.2 d) represents a very nice example of this, with a minimal size of the ellipsoid of initial well-defined states).

Our response:

1) We agree with the referee that the above mentioned statement can be misleading and that $|\kappa(t)| < 1$ due to initial system-environment correlations [as already before discussed in the 2nd paragraph below Eq. (5)]. To avoid the confusion with the text, we did the following modifications:

-We have removed the following sentence below Eq. (5):

“Since $\kappa(t)$ is independent of C_H and C_V , and the map is linear, the solution of the dynamics holds for all initial pure and mixed reduced polarization states.”

-In the 2nd paragraph below Eq. (5), where the associated point was also discussed, we have modified the text so that it reads now

“Due to initial system-environment correlations induced by the non-trivial distribution $\theta(\omega)$, we have cases with $|\kappa(0)| < 1$, i.e., the simulator uses restricted domain of initial polarization states. To simulate the channel in Eq. (1) and its decoherence function $D(t)$ with the simulator function $\kappa(t)$ in Eq.(5), we need to use the scaling $|D(t)| = |\kappa(t)|/|\kappa(0)|$.”

Referee:

2) I think that something more specific should be said about the possibility to extend the present platform to the multi-qubit case, also due to the comparison with other settings used in the literature (see, e.g., [6,7]). The authors briefly mention something about this at the end of the first column of page 6, but this seems to refer only to the “trivial” situation where each qubit is treated independently. Could it be possible to do something also considering entangled qubits, possibly even simulating an interaction among them? For example, could one exploit for this both the photons generated by SPDC?

Our response:

Indeed, we were very brief on this point in the previous version of the manuscript. To improve the presentation, we have

-Removed the sentence “It is also worth noting that the current framework can be extended to multi-qubit case by using the engineering scheme individually for each of the qubits.” in the last paragraph of the paper starting with “Summarizing...”.

and instead have added the following new paragraph just before the Summary paragraph [with new references (29-31)]:

“Current framework can be extended to multi-qubit case by using the presented dephasing engineering scheme for each of the qubits. By using both of the SPDC photons, also initial correlations between the environments (frequencies) of the qubits can be controlled to a certain extent allowing to combine dephasing control with nonlocal features of the dynamical map [29]. Moreover, it is also possible to include coherent operations to the existing set-up allowing for the exploitation of the combination of sequences of coherent operations and controllable decoherent operations, in a many-body scenario, for quantum control and simulation purposes (see also [6,7]). Lastly, activities using structured light have increased significantly during the recent years [30] including also photonic experiments [31]. Here, our results open the possibility to combine interferometry with fully controllable noise and structured photons.”

REVIEWER #2

In the manuscript NCOMMS-18-08713, the authors report on the theoretical- experimental demonstration of an optical quantum simulator of single-qubit dynamics under the influence of pure dephasing. As prototype system the authors consider a single photon with fully controlled polarization and frequency distributions.

In essence, the authors exploit the fact that the reduced density matrix describing a single photon in the polarization-frequency domain exhibits a modulation function in the coherence terms. Then, by comparing the single-photon density matrix to a generic single-qubit dynamical map under the influence of pure-dephasing, the authors elucidate that both matrices exhibit the same structure. As a result, the modulating function for the single-photon coherences can be mapped to the decoherence function of the qubit.

In the single-photon context, the “decoherence function” is the Fourier transform of the initial frequency distributions times a frequency-dependent-phase-factor for the horizontal polarization component. Hence, by tailoring the initial frequency distribution of the single photon the authors are able to emulate the dynamics of a single-qubit under the influence of pure dephasing. In short, I find this idea simple and elegant.

The paper is extremely interesting and well written. The theoretical and experimental approaches are consistent and the agreement in the outcomes is impressive. In my opinion the results are of general relevance, so that the manuscript match the level of excellence required by the Journal.

I only have a comment.

In the title the authors mention “implementation of synthetic spectral densities”. However, only in the last paragraph of the manuscript they stress “it is possible to produce synthetic spectral densities”. I would encourage the authors to provide experimental evidence of this statement. After that, I think the manuscript would be appropriate for publication in Nature Communications.

Our response:

To comply with the referee's comment, we have extended the text in the mentioned paragraph (now the 3rd last paragraph of the paper) and added a new figure (Figure 4). The added figure shows both the theoretical and experimental decoherence function dynamics of the qubit interacting with bosonic modes, the system already discussed in this paragraph earlier, and its chosen synthetic spectral density. The added text gives some more details and reads

“Ohmic spectral density is often used for dephasing dynamics. However, as an example of synthetic spectral density, we choose the one shown in Fig. 4 (a) which produces decoherence dynamics displayed both theoretically and experimentally in Fig. 4 (b). The theoretical dynamics is obtained numerically from Eq. (6) by using zero-temperature environment and the $J(\omega)$ displayed in Fig. 4(a). The experimental result has been obtained by using the frequency distribution $|g(\omega)|^2$ of the simulator photon displayed in Fig. 3 (a) while the used initial phase distribution $\theta(\omega)$ is, instead of Fig. 3 (e), a constant function. This result gives experimental evidence on the realizability of arbitrary synthetic spectral densities which we plan to study in more detail in the future.”

REVIEWER #3

Referee:

This manuscript describes the implementation of generic dephasing maps using spectral manipulation of single photons, coding polarization qubits. By imparting different spectral phases to Horizontal and Vertical components, the authors were able to replicate the effects of the coupling of a single qubit to an Ising spin chain with varying transverse field.

The work is undoubtedly interesting to specialists, but it lacks the elements of novelty that should be expected for Nature Communications. Many instances of optical quantum simulations have appeared in the last years, and, while this is an elegant demonstration, it has the same drawbacks as previous demonstrations, i.e. it is a neat way to simulate the output wavefunction, rather than directly mimicking the dynamics. While the former is still a relevant effort, the latter is more complex, hence appealing.

In addition, the experimental technique is applied for the first time to quantum simulation, however, a 4f-line it is a standard way to manipulate the spectrum by linear optical means, however much I can appreciate the ingenuity of the solution.

I therefore encourage the authors to seek publication in a prominent physics journal, which looks like a more appropriate choice.

Our response:

We would like to emphasize that our simulator is able to mimic practically any open system dephasing for a qubit - a feat that has not been achieved before to the best of our knowledge. Moreover, during the last ten years there has been large amount of activities dealing with non-Markovian open system dynamics and our results also go clearly beyond the previous achievements here. It is also worth keeping in mind that once the initial state is prepared, the set-up contains genuine time-evolution. Therefore, we think that the essence of our simulator is not simply the production of the “output wavefunction” but instead the mimicking of the open system dynamics. In general and as far as we see it, the simulator for open system should produce faithfully its dynamics whilst the properties of the environment within the simulator can be constructed and engineered for this purpose. See also Refs. (6-7) where the environment of the many-body open system simulator and its engineering includes auxiliary trapped ion.

Note also that the set-up contains the dynamics and engineering of single photon, manipulating its initial frequency distribution, and initial polarization and frequency dependent quantum mechanical phase factor. Thereby we are not quite sure what the referee means, in the context of our work, by the statement “however, a 4f-line it is a standard way to manipulate the spectrum by linear optical means, however much I can appreciate the ingenuity of the solution.”

Referee:

The authors might want to consider the following points:

- the discussion on the impact of the limited resolution of the grating and SLM system on the simulation is quite terse. Can the authors comment clearly on what features can not be replicated due to this imperfection, e.g. in terms of a frequency cut-off in the noise spectrum?

- the bibliography on the use of SLMs for qubit and single-photon manipulation and, in general, on optical simulation is very limited. Please consider extensions.”

Our response:

It is certainly true that the resolution of the SLM is limited. However, as the experimental results demonstrate, the error bars in the measurements are small, see Fig. 2 (e-h). Note also that the number of pixels used (900) is almost an order of magnitude higher than, e.g., in a recent paper (100) Ref. (8) also using SLM. As we already mention in the manuscript, some errors appear in the very short times (page 5, right column, middle paragraph). Nevertheless this does not prevent producing completely different types of decoherence function dynamics from fast decay with revival after long time interval [Fig.2(e)], to fast decay with no revivals [Fig. 2(f)] and even to coherence trapping [Fig. 2(g)]. Therefore we conclude that the current resolution of the SLM is sufficient to demonstrate the usability of our simulator for generic dephasing. Moreover, having already a sufficiently large number of frequency pixels, the key issue and achievement here is the very precise control of the available frequency pixels including also very precise control of the polarization and frequency dependent phase.

To comply with the referee’s comment, we have modified the sentences on page 5, right column, 2nd paragraph, which read now

“It is evident from Fig.~3, that producing this high control of dephasing, and being able to simulate a wide variety of dynamical features, indeed requires very challenging control of the photon frequency and frequency-dependent phase distribution. This precision is exactly what allows for the accurate mimicking of the dynamics in different dephasing regimes, even though the number of pixels of the SLM is large but still limited.”

To extend the bibliography on the use of SLMs, we have added the following sentence to the last paragraph on page 4:

“Note that SLMs have been recently used also for quantum computing and information purposes, see, e.g., Refs. [25-27].”

which includes new references [25-27].

Other changes:

-We have added the following data availability statement to the end of the manuscript, as requested by the general policy of Nature Communication:

“The data that support the findings of this study are available from the authors upon reasonable request.”

Reviewers' comments:

Reviewer #1 (Remarks to the Author):

I thank the authors for their reply and for answering to my previous remarks in a fully satisfactory way. Hence, I do recommend the paper for publication in Nature Communications.

Reviewer #2 (Remarks to the Author):

Report for manuscript NCOMMS-18-08713, "Experimental implementation of fully controlled dephasing dynamics and synthetic spectral densities"

I am quite happy with the response given by the authors and their revisions to the manuscript. At this point I have no objection to recommend the paper for publication in Nature Communications.

Reviewer #3 (Remarks to the Author):

Following previous correspondence, the authors have considered my punctual comments, however I am afraid the responses that they provide are rather unsatisfactory. I consider their replies in some detail.

"Therefore, we think that the essence of our simulator is not simply the production of the "output wavefunction" but instead the mimicking of the open system dynamics."

It does not. In the proposed scheme, the evolution equation should be solved so that its solution can be cast as a frequency-dependent phase factor on a qubit. This is not equivalent to put the qubit in contact with some device directly simulating *the coupling* (not its expected effect) and verifying that the dynamics is actually in the predicted form. As I said in my previous report, this is still a valuable effort, but less ambitious than that of a full quantum simulation.

"Note also that the set-up contains the dynamics and engineering of single photon, manipulating its initial frequency distribution, and initial polarization and frequency dependent quantum mechanical phase factor. Thereby we are not quite sure what the referee means, in the context of our work, by the statement "however, a 4f-line it is a standard way to manipulate the spectrum by linear optical means, however much I can appreciate the ingenuity of the solution." "

What I mean is that the experiment manipulates the first-order coherence of the field, as it can be, and has been in different contexts, performed on classical high-intensity fields. The simulation would give exactly the same statistics if it were conducted with coherent states - the use of single photons adds an element of elegance to it, but no substantial difference. This 4f line arrangement is widely used to shape ultrafast pulses, see J. Phys. B 43, 103001 for a review. Applying the technique to quantum state engineering is interesting, but not revolutionary. I might add concerns on the scaling of this technique to a larger number of qubits. Progresses in frequency engineering of multi-photon states, however, are promising, and could mitigate these limitations.

"the error bars in the measurements are small"

These have nothing to do with the resolution, which would lead to *systematic* errors, rather than statistic uncertainties. I expect these are linked mostly to the counting statistics.

"As we already mention in the manuscript, some errors appear in the very short times."

"Therefore we conclude that the current resolution of the SLM is sufficient to demonstrate the usability of our simulator for generic dephasing."

I have troubles understanding what the authors mean here: either the resolution of the SLM is sufficient for generic evolutions, or it produces errors. My question was indeed extremely simple: "how far can you go with this device?" I think that any paper presenting a technique should discuss features and limitations as well. It would also seem that, given the high pixel density of the SLM, the limiting factor might actually be the number of lines/mm of the grating, as well as the physical size of the beam being diffracted. This manuscript should not be considered complete without such a discussion.

We thank all the referees for their comments, suggestions and recommendations. Below, we give our remaining responses and remarks.

REVIEWER #1

Referee:

I thank the authors for their reply and for answering to my previous remarks in a fully satisfactory way. Hence, I do recommend the paper for publication in Nature Communications.

Our response:

We are happy to see that the reviewer considered our responses satisfactory and recommends the publication of the paper in Nature Communications.

REVIEWER #2

Referee:

I am quite happy with the response given by the authors and their revisions to the manuscript. At this point I have no objection to recommend the paper for publication in Nature Communications.

Our response:

We appreciate the reviewer's view on our response and the statement about recommendation for publication.

REVIEWER #3

Referee:

Following previous correspondence, the authors have considered my punctual comments, however I am afraid the responses that they provide are rather unsatisfactory. I consider their replies in some detail.

"Therefore, we think that the essence of our simulator is not simply the production of the "output wavefunction" but instead the mimicking of the open system dynamics."

It does not. In the proposed scheme, the evolution equation should be solved so that its solution can be cast as a frequency-dependent phase factor on a qubit. This is not equivalent to put the qubit in contact with some device directly simulating *the coupling* (not its expected effect) and verifying that the dynamics is actually in the predicted form. As I said in my previous report, this is still a valuable effort, but less ambitious than that of a full quantum simulation.

Our response:

The referee addresses a very subtle but important point about what it means to simulate open system dynamics. He/she describes the difference between two ways of realising an "open quantum system simulator": a) by fixing the coupling between system and environment and then letting the system evolve, and b) by simulating the "solution" of the master equation, that is a specific class of time evolutions. In the latter case b), simulating the open system dynamics obviously, by definition, requires to specify the dynamics to simulate, namely the solution of the master equation or, equivalently, the dynamical map. In case a), instead, this is not required and one only would need to engineer the *coupling*. More precisely, one would need to engineer the so called "spectral density" which tells the strength of the coupling between the system (qubit in our case) and each mode (having frequency ω) of the environment.

The referee then continues by stating that what we do is to implement a simulation of type b) which, he/she claims, is still valuable but not as much as type a).

For generic open quantum systems, specifying the spectral density, i.e., the coupling with the environment, is NOT equivalent to specifying the analytic expression of the dynamical map (solution of the master equation). In fact, in general, one may not even be able to solve analytically the master equation, and have a closed analytical form of the dynamical map. However, the case of pure dephasing dynamics is different because specifying the spectral density uniquely fixes the analytical form of the solution since the decoherence coefficient (off-diagonal term of the density matrix) only depends on the spectral density. This is true for all microscopic models leading to pure dephasing master equations, including our implementation. Our implementation is somewhat more general than, for example, the microscopic model of Ref. [18], since the coupling with the environment depends on the state of the qubit [the phase factor $\theta(\omega)$ controlling the dynamics appears only in the H polarisation part of the initial state]. This gives us additional flexibility. As one can see from Eqs. (2), (4) and (5) of our manuscript, specifying the complex spectral density [$g(\omega)$ and $\theta(\omega)$], i.e., specifying the *coupling* between system and environment, uniquely determines the dynamical map via Eq. (5). So open quantum system simulations by fixing the coupling as in a) or fixing the dynamics/solution as in b), for pure dephasing models, are analogous to each other.

For the specific example considered in our manuscript to illustrate the behaviour of the simulator, i.e., a central spin coupled to an Ising chain in a transverse field, what we know explicitly from previous literature (Refs [19] and [29]) is NOT the spectral density but the dynamics of the central spin, namely the time evolution of the density matrix of the central spin coupled to the Ising chain. Hence we need to extract, first, the corresponding spectral density and then simulate the dynamics by synthesizing such spectral density. This is precisely what we do in our manuscript. However, in general, we stress once more that for single qubit dephasing, knowing $g(\omega)$ and $\theta(\omega)$ is equivalent to knowing the time evolution of the density matrix, hence engineering the coupling by controlling the spectral density and engineering the dynamics amount to similar thing.

To comply with the reviewer's comment, we have added the following text in the end of the paragraph below Eq.(5):

"For generic open quantum systems, specifying the spectral density (i.e., the coupling with the environment) is not equivalent to specifying the analytic expression of the dynamical map (solution of the master equation). In fact, in general, one may not even be able to solve analytically the master equation, and have a closed analytical form of the dynamical map. However, the case of pure dephasing dynamics is different because specifying the spectral density uniquely fixes the analytical form of the solution since the decoherence function (off-diagonal term of the density matrix) only depends on the spectral density, see Eqs. (2), (4) and (5)."

Referee:

"Note also that the set-up contains the dynamics and engineering of single photon, manipulating its initial frequency distribution, and initial polarization and frequency dependent quantum mechanical phase factor. Thereby we are not quite sure what the referee means, in the context of our work, by the statement "however, a 4f-line it is a standard way to manipulate the spectrum by linear optical means, however much I can appreciate the ingenuity of the solution." "

What I mean is that the experiment manipulates the first-order coherence of the field, as it can be, and has been in different contexts, performed on classical high-intensity fields. The simulation would give exactly the same statistics if it were conducted with coherent states - the use of single photons adds an element of elegance to it, but no substantial difference. This 4f line arrangement is widely used to shape ultrafast pulses, see J. Phys. B 43, 103001 for a review. Applying the technique to quantum state engineering is interesting, but not revolutionary. I might add concerns

on the scaling of this technique to a larger number of qubits. Progresses in frequency engineering of multi-photon states, however, are promising, and could mitigate these limitations.

Our response:

We thank the referee for the detailed explanation, a 4f-line is indeed a standard way to manipulate the spectrum by linear optical means. We misunderstood this point in the previous response and modifications. As the referee points out, we apply the simultaneous amplitude and phase modulation of spectrum to quantum state engineering, and subsequently achieve the controlled dephasing dynamics of a qubit in the experiment. The frequency engineering, and in general pulse shaping of multi-photon states, is indeed interesting area of research, and we plan to look for connections between our approach motivated on open system dynamics and multi-photon pulse shaping in the future. To comply with the referee's comment, we have added the following text towards the end of page 5 with new reference [28] given by the referee. The full sentence now reads (new text is the part concerning the 4f-line):

"Note that SLMs have been recently used also for quantum computing and information purposes, see, e.g., Refs. [25-27], and that a 4f-line is a standard way to manipulate the spectrum and implement pulse shaping by optical means [28]."

Referee:

"the error bars in the measurements are small"

These have nothing to do with the resolution, which would lead to *systematic* errors, rather than statistic uncertainties. I expect these are linked mostly to the counting statistics.

"As we already mention in the manuscript, some errors appear in the very short times."

"Therefore we conclude that the current resolution of the SLM is sufficient to demonstrate the usability of our simulator for generic dephasing."

I have troubles understanding what the authors mean here: either the resolution of the SLM is sufficient for generic evolutions, or it produces errors. My question was indeed extremely simple: "how far can you go with this device?" I think that any paper presenting a technique should discuss features and limitations as well. It would also seem that, given the high pixel density of the SLM, the limiting factor might actually be the number of lines/mm of the grating, as well as the physical size of the beam being diffracted. This manuscript should not be considered complete without such a discussion.

Our response:

We thank the referee for the comments and agree that the discussion on systematic errors was previously not sufficient.

To comply with the referee's comment, and to add the required discussion on features, limitations and sources of systematic errors, we have added three new paragraphs.

The old text on page 5

"Some errors in the results appear in the very short time regime when simulating the paramagnetic phase and the phase transition points. This is due to technical limitation and finite number of pixels in the SLM (900) but the accuracy could be improved by increasing the number of pixels. Also, due to the nature of the Fourier transformation, largest errors appear in principle at very short and long times. The error bars, when simulating the ferromagnetic phase displaying coherence trapping of the qubit, increase due to the very narrow frequency distribution [c.f.~Fig. 3(c)] which also reduces the number of photon counts."

has been replaced with new text

“It is worth noting that systematic errors have a non-negligible effect. Figure 2(g) is a typical example of the resolution that leads to these errors. The corresponding hologram is in Fig. 1 (inset iii). It becomes more difficult to modulate the amplitude of the frequency with high fidelity, when the spectrum gets narrower (also beam mode becomes worse). Although the setup was carefully optimized, these factors still lead to decrease in the fidelity of the initial state preparation. This is the reason why the experimental result [Fig. 2(g)] does not show an agreement with the theory [Fig. 2(c)] as good as in the other figures.

Having too wide spectrum also leads to systematic errors. Fig. 2(a), (b), and (c) display the dynamics of the decoherence function for the spin-system parameters $\lambda = 0.01$, $\lambda = 0.9$ and $\lambda = 1.8$, respectively, using 4000 spins in the environment. Although three gratings (1200 l/mm) are used, 3 nm FWHM (full width at half maximum) of SPDC photons can only cover 900 pixels in SLM. This effectively means that we can simulate 900 out of all 4000 environmental spins. 3100 spins corresponding to amplitude and phase close to 0 are ignored in the set-up. The experimental results [Fig. 2(e), (f), and (g)] are slightly different with respect to the theoretical results [Fig. 2(a), (b) and (c)]. In principle, this systematic error can be reduced by using smaller pixel SLM and wider FWHM filter. Note that, unless the spectrum is too narrow, the beam mode is good enough to achieve simulation with high fidelity.

For gratings, three 1200 l/mm gratings are used in our setup. This is a result of tradeoff. 1800 l/mm grating will make the divergence of spectrum bigger, but the fidelity of polarization states will be significantly less than the fidelity with gratings of 1200 l/mm. Therefore, increasing the divergence angle of the spectrum by increasing the density of the grating is somewhat challenging.”

OTHER CHANGES:

-As required by Nature Communications format, we have added headings for the sections Introduction, Results and Conclusions. Section Results also contain now the added subheadings.

-As required by Nature Communications format, we have added the figure captions also to the end of the manuscript.

REVIEWERS' COMMENTS:

Reviewer #3 (Remarks to the Author):

The authors have implemented the necessary changes to make their manuscript technically correct, and clarified its positioning with respect existing literature. As to whether its subject is suitable for Nature Communications, it is up to the editor to decide.